Tunable translational control using site-specific unnatural amino acid incorporation in Escherichia coli

Kato Yusuke kato@affrc.go.jp
Genetically Modified Organism Research Center, National Institute of Agrobiological Sciences , Tsukuba, Ibaraki , Japan
Gatti-Lafranconi Pietro
Electronic publication date: 2015 Apr 28
Publication date: 2015
Volume: 3
Electronic Location ID: e904
Received 2014 Dec 12; Accepted 2015 Mar 31
Copyright: © 2015 Kato
Copyright year: 2015
Copyright holder: Kato
License: This is an open access article distributed under the terms of the Creative Commons Attribution License, which permits unrestricted use, distribution, reproduction and adaptation in any medium and for any purpose provided that it is properly attributed. For attribution, the original author(s), title, publication source (PeerJ) and either DOI or URL of the article must be cited.
License URL: https://creativecommons.org/licenses/by/4.0/

Keywords: Translational regulation, Escherichia coli, Expression system

Funding: JSPS KAKENHI 25660281 This work was supported by JSPS KAKENHI Grant number 25660281. The funders had no role in study design, data collection and analysis, decision to publish, or preparation of the manuscript.

==============================
Translation of target gene transcripts in Escherichia coli harboring UAG amber stop codons can be switched on by the amber-codon-specific incorporation of an exogenously supplied unnatural amino acid, 3-iodo-L-tyrosine. Here, we report that this translational switch can control the translational efficiency at any intermediate magnitude by adjustment of the 3-iodo-L-tyrosine concentration in the medium, as a tunable translational controller. The translational efficiency of a target gene reached maximum levels with 10−5 M 3-iodo-L-tyrosine, and intermediate levels were observed with suboptimal concentrations (approximately spanning a 2-log10 concentration range, 10−7–10−5 M). Such intermediate-level expression was also confirmed in individual bacteria.

Introduction

Controlling gene expression is a key methodology for biotechnology. Particularly, its fine-tuning and optimization are important for construction of artificial gene circuits in synthetic biology and for metabolic engineering. Tunable conditional expression systems regulated by extracellular inducers/repressors are thus very useful.

We previously demonstrated a translational switch using site-specific unnatural amino acid (UAA) 3-iodo-L-tyrosine (IY) incorporation in natural amber suppressor-free strains of Escherichia coli (Minaba & Kato, 2014). Although IY is not perfectly “unnatural” and found in very specific biological tissues such as the thyroid cells (Tietze et al., 1989) and the sponge skeletons (Ueberlein et al., 2014), we cannot detect IY ubiquitously in natural environments. The translational switch is based on conditional read-through of the UAG amber stop codons that are inserted in target genes (Fig. 1). A variant of aminoacyl-tRNA synthetase (aaRS) IYRS that was derived from the archaebacterium Methanocaldococcus jannaschii specifically recognizes both IY and an amber suppressor tRNA (tRNACUA) MJR1 (Sakamoto et al., 2009). Extracellular IY is taken up and incorporated into proteins at sites of the amber codons in the IYRS/MJR1-expressing cells. The target gene transcripts with the amber stop codons inserted next to the AUG translational start codon are translated only in the presence of IY. The absence of IY prevents translation of the target genes.

Figure 1 Schematic of the translational switch using the amber codon-specific IY.

An amber stop codon is inserted next to the ATG translational start codon in the target gene (egfp). MJR1 is an amber suppressor tRNA. IYRS is an aminoacyl-tRNA synthetase that orthogonally recognizes IY and MJR1. Extracellular IY is taken up by the bacteria. The addition of IY in the media results in amber stop codon read-through and translation of the target gene. Translation is interrupted in the absence of IY. RF1, peptide chain release factor 1.

The UAA-controlling translational switch has distinct features for gene regulation (Minaba & Kato, 2014). First, this translational switch also can be controlled by the induction/repression of either aaRS or tRNACUA, in addition to the presence/absence of UAA. The switchability can be modulated by combination use with aaRS- and/or tRNACUA- switching. Second, UAA does not naturally exist. Third, UAA is a “building block” that forms an aminoacyl-tRNA and target proteins, distinct from simple regulation-specific molecules, such as isopropyl β-D-1-thiogalactopyranoside (IPTG) for the derepression of the lactose operator. The switching mechanism involves the direct incorporation of UAA into target proteins, and is not effective for interventional systems. The second and third characteristics indicate that the UAA-controlling translational switch is robust against environmental and host-endogenous noises. Although an inducible tRNACUA has been used as a similar translational switch based on amber suppression, these features are not found (Zengel & Lindahl, 1981; Herring, Glasner & Blattner, 2003). Fourth, the UAA-controlling translational switch can be used in combination with any established transcriptional switches to obtain a synergistic regulatory effect. Using this strategy, we constructed a high-yield and zero-leakage expression system, in which strong expression by the T7 promoter was maintained under the induction condition, and almost no proteins were expressed under the repression condition by double control of transcription-translation.

Riboswitches and small regulatory RNAs (sRNAs) are well known as post-transcriptional controlling tools in E. coli. Some riboswitches that are located in non-coding portions of mRNAs can regulate gene expression in cis by binding a specific small molecule via controlling translational initiation and/or mRNA degradation (Caron et al., 2012). The sRNAs are trans-acting and target gene alterations are not required, although off-target effects are often detected (Bobrovskyy & Vanderpool, 2013). The sRNAs modulate the translation via controlling translation initiation and/or mRNA degradation. The distinct characteristics of these RNA-based post-transcriptional switches suggest that the UAA-controlling translational switch is complementary rather than competitive with the others.

To date, many systems for the incorporation of various UAAs have been developed (Liu & Schultz, 2010). In addition, similar translational switches controlled by UAAs are expected to be applicable also for eukaryotes, such as yeasts, nematodes, insects, mammalian cells, and plants, because the site-specific unnatural amino acid incorporation systems have already been introduced (Chin et al., 2003; Greiss & Chin, 2011; Bianco et al., 2012; Sakamoto et al., 2002; Li et al., 2013).

The target gene products that are controlled by the UAA-controlling translational switch necessitate the incorporation of UAA. Although the UAA-incorporation may cause functional alterations in some target proteins, we can avoid this problem by neutral site selection, incorporation into tag or processed-out sequences. Alternatively, the UAAs can be used as tools to facilitate purification or tracking of target gene products (Minaba & Kato, 2014).

In previous studies, we only focused on the on-off aspect of the IY-controlling translational switch. Here, we studied the intermediate states between the fully on- and off-states of the translational switch. The results suggest that this switch can control the translational efficiency at any intermediate magnitude by adjustment of the appropriate IY concentration, and thus, function as a tunable translational controller.

Materials and Methods

Fluorescence measurement of pooled bacteria

Assays were performed as described previously (Minaba & Kato, 2014). Briefly, we used E. coli BL21-AI (F−ompT gal dcm lon hsdSBrB−mB−) araB::T7RNAP-tetA) carrying the plasmid pTYR MjIYRS2-1(D286) MJR1 × 3 encoding IYRS and MJR1 and the amber-inserted EGFP expression plasmid (driven by the E. coli lpp promoter with an amber stop codon TAG inserted next to the start codon ATG). The sequence for the EGFP expression plasmid is shown in Fig. S1. An overnight (approximately 16 h) culture of bacteria was resuspended in an equal volume of liquid LB medium (1% bacto tryptone, 0.5% yeast extract, and 1% NaCl) containing various concentrations of optically-pure IY. IY was directly dissolved and diluted in LB medium. After a 6 h culture (2 ml in a 14 ml-culture tube at 37 °C with rotary shaking at 200 rpm), the bacteria were collected by centrifugation (1,800 × g for 3 min). The pellet was washed and was resuspended in an equal volume of 0.9% (wt/vol) NaCl, and this wash step was repeated twice. An aliquot of the bacterial suspension (150 µl) was diluted in 3.0 ml of the 0.9% NaCl, and the OD590 was measured. The fluorescence intensity of the bacterial population was measured using a Shimadzu RF-5300PC spectrofluorometer (excitation, 480 nm; emission, 515 nm). The background fluorescence was estimated using the bacteria carrying a MJR1-deleted plasmid (ΔMJR1) at IY = 0 (no significant IY-dependency was observed, Fig. S2). The background-subtracted values were used to calculate the points for the dose–response curve. For the time course measurement of EGFP accumulation, we cultured the bacteria in 20 ml LB medium in a 200 ml culture flask. Aliquots of the bacterial culture (150 µl) were withdrawn at various time points, and the OD590 and fluorescence were measured.

Photomicrographs

Photomicrographs and Nomarski differential interference contrast images of the fluorescent bacteria were recorded using both a Carl Zeiss Axioskop 2 with a 38-HE Endow GFP filter-set (Carl Zeiss, Jena, Germany) and a Roper Scientific Photometrics CoolSNAP ver.1.1 (Roper Industries, Sarasota, Florida, USA).

Image analyses

Analyses of the fluorescence images were performed using ImageJ 1.48v (National Institutes of Health, Bethesda, Maryland, USA). Prior to analyses, the fluorescence images were confirmed as not being saturated at any pixels. Three-dimensional graphs of the intensities of the pixels were generated using Surface Plot command. The fluorescence intensity of individual bacteria was measured using Particle Analysis command. A background value (bacteria-absent area) was used as a threshold for particle detection. A range of particle area was set to detect only individual and not-overlapping bacterial cell images.

Growth curve

Growth curves were determined as previously described with some modifications (Minaba & Kato, 2014). Overnight cultures of bacteria were diluted (1:200) in fresh LB medium and were incubated at 37 °C with rotary shaking at 200 rpm. After reaching a visible density (around OD590 = 0.1), OD590 was measured every 20 min for 2 h.

Results and Discussion

To characterize the intermediate states of the translational switch using the IY-incorporation system, the IY dose-dependency of translational efficiency for a target gene was first determined for a bacterial population (Fig. 2A). As shown in Fig. 1, an EGFP gene containing an amber stop codon next to the ATG translational start codon was constitutively transcribed by the lpp promoter in the E. coli BL21-AI strain expressing IYRS and MJR1. The translational efficiency was estimated from the EGFP fluorescence 6 h after IY addition into the medium. The “gross” translational efficiency for the population started to increase significantly at 3 × 10−7 M and reached maximum levels at 10−5 M. Intermediate levels of translational efficiency were observed at a 2-log10 suboptimal concentration of IY (10−7–10−5 M), suggesting that the translational efficiency can be tuned in this concentration range. The 50% effective concentration was estimated to be 3 × 10−6 M. The IY concentration-dependent change of the gross translational efficiency was also confirmed by a direct measurement of the time course for EGFP accumulation (Fig. 2B). The EGFP accumulation rate was clearly slower at suboptimal concentrations than that at the optimal concentration. The accumulation rate increased with increasing IY concentration, also suggesting that an intermediate translational efficiency could be obtained in the suboptimal concentration range.

Figure 2 IY dose-dependent change in translational efficiency for a bacterial population.

The experimental system is schematically shown in Fig. 1 (A) Dose-response curve. Bacteria were cultured in media containing various concentrations of IY for 6 h. Colored circles indicate the samples for Fig. 3. Data are shown as mean ± SEM. n = 3 independent experiments using completely separated bacterial cultures. Statistical analysis was performed using Welch’s t-test in Excel ver. 14.0. A single fitted dose-response curve (log-logistic, IY = 0–3 × 10−4 M) was generated using Origin7. The equation for the fitted curve and assessments of goodness-of-fit are represented under the graph. F, fluorescent intensity (arbitrary unit); Fmax, the initial F value (right horizontal asymptote); Fmin, the final F value (left horizontal asymptote); CIY, IY concentration (M); C0, 50% effective CIY (point of inflection); χ2, reduced chi-squared value; DoF, degrees of freedom. (B) Time course of EGFP accumulation.

Intermediate levels of gene expression for the pooled bacteria are not always equal to that for an individual bacterium. In the case of conditional gene expression by the araBAD promoter, intermediate expression levels in the cultures reflected a population average of the induced and non-induced cells, i.e., each cell responded to an inducer L-arabinose at a suboptimal concentration in an all-or-none manner (Siegele & Hu, 1997; Guzman et al., 1995). A similar non-uniform induction was also reported for the lac operon (Novick & Weiner, 1957; Maloney & Rotman, 1973). We therefore determined whether the IY dose-dependent change of EGFP fluorescence in individual cells using fluorescence images (Fig. 3). The fluorescence intensity of these images was quantified by image analysis, and spatial maps were generated. The fluorescence intensity clearly increased with increasing IY concentration. Fluorescent intensity distribution histograms of individual bacteria were also examined. As expected from the fluorescence images and their spatial maps, the peak frequency at a suboptimal IY concentration (1 × 10−6 M) was located between those at zero and the saturation concentration, indicating that the translational efficiency of individual bacteria can be controlled at intermediate values. This response is unlike the mix-population of all-or-none responding cells observed in a suboptimal concentration of inducers for both the araBAD promoter and the lac operon. The frequency distribution was relatively wide and overlapped with those at zero and at the saturation concentration, suggesting that the responses of individual cells are variable at suboptimal concentrations. The all-or-none behavior of the lac operon was explained by an autocatalytic positive feedback loop that led to a burst of synthesis of galactoside permease LacY (Novick & Weiner, 1957; Maloney & Rotman, 1973). A similar model was also proposed for the araBAD promoter system (Siegele & Hu, 1997). In these models, the intracellular concentration of either lactose or arabinose tends to be either very low or saturated. To obtain a better linear response to an extracellular inducer concentration, IPTG is used instead of lactose in a lacY− strain for control of the lac promoter (Khlebnikov & Keasling, 2002). Similarly, mutant strains in which arabinose transport and degradation genes are deficient can avoid the all-or-none response of the araBAD promoter (Bowers et al., 2004). These modifications impair the positive feedback loop and confer linearity between extracellular and intracellular inducer concentrations. In the case of the IY-controlling translational switch, the velocity of translation by a suppressor IY-tRNA is expected to depend on the intracellular IY concentration if the concentrations of the suppressor tRNA, IYRS, and peptide release factors are constant (Yarus et al., 1986). The intracellular IY concentration was possibly at a subsaturation level at the suboptimal concentration of extracellular IY. Although the uptake mechanism of IY remains unclear, positive feedback loops as seen in the lac operon or the araBAD system may be weak or not be involved (Pittard, 1996).

Figure 3 EGFP expression in individual bacteria.

The sampling points are indicated in the dose-response curve in Fig. 2A. Upper and lower photographs are epifluorescence images and Nomarski differential interference contrast images, respectively. The photographic conditions were constant for all of the fluorescence images. Calibration bar = 100 µm. Asterisks in spatial distribution graphs indicate the upper right corners of the fluorescence images. The frequency in the histograms indicates the number of individual bacteria.

Approximately 7% of the maximum translation was detected even in the absence of IY, as also described previously (Minaba & Kato, 2014). The leaky translation was completely abolished by deletion of the MJR1 gene, suggesting that mischarges of MJR1 were the cause (Figs. 4A and 4B). Although some countermeasure techniques have been proposed, reduction of leaky translation needs to be a priority (Minaba & Kato, 2014).

Figure 4 Leaky translation in the absence of IY.

(A) Leaky translation for a bacterial population. Note that the measured fluorescence contains both EGFP-fluorescence and non-EGFP background. Complete set, the strain carrying the plasmid pTYR MjIYRS2-1(D286) MJR1 × 3 and the amber-inserted EGFP expression plasmid (driven by the E. coli lpp promoter); ΔMJR1, a derivative strain carrying a MJR1-deleted plasmid; ΔEGFP, a derivative strain lacking the amber-inserted EGFP gene expression plasmid; ΔEGFP + Lux, a derivative strain in which the amber–inserted EGFP gene was substituted with an amber-inserted LuxB gene from the bacterium Vibrio harveyi as a vector control (Schultz & Yarus, 1990). Data are shown as mean ± SEM. n = 3 independent experiments. Statistical analysis was performed using Welch’s t-test. ns, not significant. (B) Observation of leaky translation in individual bacteria. The EGFP fluorescence was measured in the absence of IY. In these images, non-specific background fluorescence was completely filtered out (note that this method is distinct from that of Fig. 4A). The exposure time for the EGFP images was twice that in Fig. 3.

The translational efficiency decreased from the saturation level at an extremely high IY concentration (Figs. 2A and 2B). Although its mechanism remains to be elucidated, the decrease may not be due to non-specific toxicity of IY because the bacterial growth rates did not decrease both in the IY-incorporating strains and in the parent strain (Fig. S3).

In this study, we evaluated the IY-controlling translational switch using a single setting (a single amber codon in one position, and the single target gene EGFP). In the future, further studies using this application should clarify how general this system is.

Conclusions

The translational switch using site-specific IY incorporation can be used as a “tunable translational controller” that regulates the translational efficiency in each individual cell. Using this controller, we expect to be able regulate the translational efficiency over a wide range in combination with any transcriptional controlling systems (Minaba & Kato, 2014). The tunable translational controller is a promising tool for conditional fine-tuning and for optimizing the construction of artificial gene circuits in synthetic biology and in metabolic engineering (Yadav et al., 2012).

Supplemental Information

Figure S1 Sequence of an amber-inserted EGFP expression construct

The amber-inserted EGFP expression sequence was cloned into pDONR221 using a standard Gateway reaction by BP recombination.

Click here for additional data file.

Figure S2 EGFP fluorescence of bacteria carrying the ΔMJR1 negative control plasmid

A bacterial strain carrying both the ΔMJR1 and the amber-inserted EGFP expression plasmid was evaluated. EGFP fluorescence was measured at various IY concentrations. Data are shown as mean ± SEM. n = 3 independent experiments. Statistical analysis was performed using Welch’s t-test (α = 0.05). No significant differences were detected in EGFP fluorescence intensity.

Click here for additional data file.

Figure S3 Growth curves at an extremely high IY concentration

(A) pTYR MjIYRS2-1(D286) MJR1 × 3 and the amber-inserted EGFP expression plasmid. (B) pTYR MjIYRS2-1(D286) MJR1 × 3 alone. (C) The parental strain BL21-AI without any plasmids. Open circle, 0 M; filled circle, 3 × 10−3 M.

Click here for additional data file.

Supplemental Information 1 Data for Fig. 2A

Click here for additional data file.

Supplemental Information 2 Data for Fig. 2B

Click here for additional data file.

Supplemental Information 3 Data for Fig. 3 (histogram).

Click here for additional data file.

Supplemental Information 4 Data for Fig. 4A

Click here for additional data file.

We thank Kensaku Sakamoto and Shigeyuki Yokoyama (RIKEN) for the IYRS-MJR1 expression plasmids and Michael Yarus (University of Colorado) for the luxB gene.

Additional Information and Declarations

Competing Interests

Author Contributions

The author declares there is no competing interests.

Yusuke Kato conceived and designed the experiments, performed the experiments, analyzed the data, contributed reagents/materials/analysis tools, wrote the paper, prepared figures and/or tables, reviewed drafts of the paper.

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
