# Peer review of "Tunable translational control using site-specific unnatural amino acid incorporation in Escherichia coli"

_PeerJ, doi:10.7717/peerj.904_

## Round 0.1 · original submission · Minor Revisions

The submitted manuscript addresses a topic of broad interest in the field and would meet the interest of the community.

As all of the reviewers have pointed out, however, some additional details are needed to support the author's claims. In particular, the following points should be prioritised in preparing the revised version:

1. Detailed Experimental Section: reviewer 1 and 2 list the key information missing from the current manuscript and that should be added

2. Background: as pointed out by reviewers 2 and 3, the provided background information is currently kept to the minimum. Additional evidence on the use and performance of other similar systems (including those not developed by the author) should be included to support the importance of the advances here presented.

3. Conclusions: stating the generality of the presented approach and its relevance (compared to other systems, see point 2 and comment from reviewer 2 and 3) would strengthen the final claims. In addition, the issue of fluorescence uniformity (reviewers 1 and 3) should be explicitly addressed.

4. Clarity can be improved throughout the manuscript by following the grammar and language suggestions given by reviewer 2.

The detailed and overall positive comments given from the three reviewers provide accurate and straightforward guidelines for the revision of the manuscript. I believe that all points made by the three reviewers are valid, and encourage the author to consider them for revision even if they are not featured in the priority list above.
I will be looking forward to reading the revised version.

Reviewer 1 ·

Basic reporting

Submission is well written in the most part and easy to understand. A couple of clarifications are needed, I think:

P4 L44 – “no molecules were expressed” – do you mean no protein was produced?

P7 L86 – “The intracellular IY concentration was possibly a subsaturation level at the suboptimal concentration of extracellular IY” – I am not sure I understand this point.

Experimental design

I think that the materials & methods section could have a few more details in, for example:
Composition of LB needs to be stated as there are multiple recipes.
What temperature were cultures grown at? What volume? What type of flask? What shaking speed? How were bacteria washed? What is IY dissolved in?
How was fluorescence measured?

Validity of the findings

Fig 2B – I am not sure I agree with the assertion that all the cells at 10-6 M IY show the same fluorescence – there also appears to be heterogeneity at zero IY. Can cellular fluorescence be quantified using image analysis, or perhaps flow cytometry? Also, the image for 3 x 10-3 M IY appears to be saturated so would be difficult to quantify – again, image analysis should be used to back up the conclusions drawn from the image data. As this is one of the main points that is being made by the paper, I think that quantification is important here.

Reviewer 2 ·

Basic reporting

see below

Experimental design

see below

Validity of the findings

see below

Additional comments

In this short report Yusuke Kato describe that the expression of EGFP site-specifically incorporating iodo-tyrosine via amber suppression can be regulated in a tunable manner by the concentration of iodo-tyrosine present in the medium. While the work is technically sound, it makes only incremental advance (concentration dependence of iodo-tyrosine) over preexisting work by the same author, where iodo-tyrosine dependent protein expression was reported earlier. Please also consider the following issues:
The materials and methods section is too brief and fails to provide several key information: a) The position of the amber codon in EGFP and the sequence of this expression construct; b) The conditions (media, temperature, etc) at which protein expression was performed; c) If stereochemically pure iodo-tyrosine was used or if it was D/L mixture; d) how many replicates of the experiments were performed for the error analyses, etc. Instead of just end point analyses, time dependent GFP expression at various iodotyrosine concentrations should also be investigated, if this is to be reported as a generally applicable method.
A fundamental issue with using unnatural amino acid (UAA) levels to control the expression of a desired protein lies in the obligatory incorporation of the UAA into the protein, thus potentially changing its properties. Also, amber suppression efficiency is notoriously context-dependent, and different sites in different proteins do not behave the same way, which makes it very difficult to use this as a general strategy for the regulation of protein expression. While this technology may still have potential utility, these limitations should be discussed in the manuscript for the sake of candor.

·

Basic reporting

There are a number of grammatical errors throughout the text. They interfere with clarity and should be revised. I list below the corrections I would suggest for the abstract and I leave the author to address the remaining of the paper:

L.16 – “with UAG amber stop codon inserts” – “harbouring UAG amber stop codons” would be better.
L.17 – “using amber-specific incorporation of… this unnatural amino acid.” – “on by the amber-codon-specific incorporation of an exogenously supplied unnatural amino acid, 3-iodo-L-tyrosine.” I believe would be an improvement.
L.18 – “found” – “report” would be more appropriate.
L.19 – I don’t understand what is meant by “non-stepwise”. Does that mean non-linear?
L.19 – “appropriate” is redundant and can be omitted.
L.21 – “maximum levels with 10-5 to 10-3 M” – The data presented support that the maximum level was at around 10-5 M.
L.22 – “and intermediate levels were… range, 10-7 to10-5 M).” – may be better described by specifying the range of the inducer concentration and potentially the equation of the best-fit.

Background is very brief. It refers only to recent research published by the author. Some discussion of tunable promoters is present in the results section but does not include more recently developed systems that are truly tunable.

Regarding Fig. 2A/B, is the concentration shown in red in 2B correctly represented in 2A?

Experimental design

I would welcome a couple of minor clarifications on the methodology.

1. EGFP expression as a function of IY concentration (Fig. 2A) – it is not clear what IY concentration was used to determine the background. Similarly, has the author checked whether the fluorescence background in the ΔMJR1 strain is independent of IY concentration?

2. The author claims that uniform expression was confirmed in individual bacteria – it is not clear how the fluorescence per cell was quantified and any observed variation analysed. The data presented in Fig. 2C are a bulk measurement and does not support that statement.

3. ANOVA requires normally distributed data and the variances between groups to be equal. Using the data in Fig. 2A, there is a clear correlation between error and the fluorescence measurement, suggesting that the data are not from a normal distribution. Could the author justify the choice of analysis?

4. Bacterial growth conditions – I would be grateful if the author could include more details on the size of the culture and growth conditions (particularly aeration conditions) used for the dose-response curve experiment.

Validity of the findings

As no data are presented on the variation of fluorescence between cells, I do not think the author can claim that fluorescence was uniform between individual cells.

The tunable system reported is based on the demonstration of the expression of one protein (EGFP) using a single amber codon in a single position of the mRNA transcript. Therefore, there is an unanswered question of how general the system is. Answering that may be beyond the scope of this report but the authors should make that limitation clearer.

Do the authors understand why fluorescence drops if IY concentrations go above 10-5 M? Similarly, how well does the 50% effective IY concentration match the Km of IY for the MjTyrRS?

Additional comments

Uniform expression is highly desirable for bioprocessing. If the IY system is general, it may be of interest to the biochemical engineering community.

---

## Round 0.2 · Minor Revisions

The revised manuscript has been improved significantly and has implemented the concerns/suggestions provided by the reviewers.
The additional comments provided by reviewer 3 are all minor but would still improve the manuscript. As they will require little time to be implemented, I expect to be approving a final version of the manuscript very soon.

Reviewer 1 ·

Basic reporting

No comments - see below.

Experimental design

No comments - see below.

Validity of the findings

No comments - see below.

Additional comments

I am happy that my comments have been answered in a satisfactory manner. I am especially happy to see that the GFP data has been quantified and is described in a more precise manner.

Reviewer 2 ·

Basic reporting

please see below

Experimental design

please see below

Validity of the findings

please see below

Additional comments

The revised version of this manuscripts has achieved significant improvement. The methods section now is clearly written, and most of the original concerns were taken into account.

·

Basic reporting

The article has improved from the previous version. I still have some criticisms but they are minor and I leave to the editors' discretion whether they should be addressed or not prior to publication.

line 48 “UAA does not naturally exist” - that may be inaccurate. I think 3-iodo-L-tyrosine is an early precursor of a thyroid hormone.

lines 45 - 55 could be edited for clarity.

line 56 (and other instances) - I think “UAA-controlled” may be preferable to “UAA-controlling”

line 60 - “Riboswitches…” should be the start of a new paragraph.

lines 74 - 76 could be edited for clarity.

line 277 - I think “Excel ver. 14.0” would suffice.

Experimental design

Fine.

Validity of the findings

M&M - The fitting of the IY dose response is not explained, the equation and the goodness of fit are not given. The fitted curve could also be added to Fig 2A. The benefit of having a fit is that one would then be able to predict translation levels for any given IY concentration - which I understood to be the aim of the study.

M&M - The Welch’s t-test is an improvement in the analysis but still assumes that the underlying distribution is normal. While I don’t think the conclusions of the paper will be affected, rigorous analysis would require transformation of the data to break the correlation between mean and variance. The transformed data (then a normal distribution) would then be suitable for a t-test or ANOVA.

---

## Round 0.3 · accepted · Accept

I thank the author for taking on board these further comments when appropriate and for argumenting their choice against a specific (and not essential) change. The manuscript is now acceptable for publication.
I am convinced the scientific community will benefit from the author's findings and attempt to rationalise experimental evidence in a predictive model.